# Impact Force Identification of the Variable Pressure Flexible Impact End-Effector in Space Debris Active Detumbling

**Ziying Wei [1], Huibo Zhang [1] , Baoshan Zhao [2], Xiaoang Liu [1],\* and Rui Ma [2],\***

[1] School of mechanical engineering, Hebei University of Technology, Tianjin 300130, China; 201921202011@stu.hebut.edu.cn (Z.W.); zhanghb@hebut.edu.cn (H.Z.)

[2] Tianjin Key Laboratory of Microgravity and Hypogravity Environment Simulation Technology, Tianjin 300000, China; z06010426@gmail.com

\* Correspondence: 2015044@hebut.edu.cn (X.L.); 201921202069@stu.hebut.edu.cn (R.M.); Tel.: +86-1329-995-3122 (X.L.); +86-1832-210-5885 (R.M.)

**Abstract:** The security of the space environment is under serious threat due to the increase in space debris in orbit. The active removal of space debris could ensure the sustainable use of the space environment; this removal relies on detumbling technology. According to the characteristics of the mechanical impact-type active detumbling method, this paper discusses a method to accurately identify the impact force using a pressure sensor. In this work, the impact force between a flexible impact end-effector and the space debris was analyzed theoretically and experimentally considering the pressure change during impact. Firstly, a nonlinear impact force model was established for the impact between a flexible end-effector and space debris. Secondly, impact experiments were performed and the friction model was modified. Finally, the effect of detumbling was verified through simulation experiments. The results showed that the identification error of normal impact force was less than 6.7% and the identification error of tangential friction force was less than 6.9%. Therefore, this identification method of impact force met the requirements of space debris detumbling, which has important guiding significance for the active removal technology of space debris.

**Keywords:** detumbling; mechanical impact; flexible end-effector; impact force

## 1. Introduction

The development of the human spaceflight industry is threatened by a huge amount of space debris, such as residual rocket stages, uncooperative satellites, disintegrated spacecraft, and collision derivatives [1]. As of 2017, the total number of space debris pieces in the Earth's orbit is in the tens of billions [2]. Large space debris, such as rocket final stages and failed satellites, weigh more than 1000 kg and rotate at speeds of more than 25°/s. Super-high-speed collisions could occur between them, which would lead to the formation of large-area high-speed debris clouds [3]. The damage caused by large space debris is huge, and they are difficult to remove. The National Aeronautics and Space Administration's statistics show that if two debris were removed each year starting from 2020, the growth trend of the number of debris would significantly reduce. If five debris were removed each year starting from 2020, the growth trend would stabilize [4]. The active removal of space debris is the only effective way to completely curb the increase in the number of debris. Therefore, the active removal of space debris is currently one of the hot topics in the field of space research.

Capturing debris in orbit is the key to active removal, however, this is quite difficult to achieve, because the motions of space debris are very complicated. Space debris are affected by such factors as space perturbation force or torque and residual angular momentum before failure, and are often

in a state of high speed and complex tumbling motion [5]. If the angular momentum of the target is reduced before the target is captured, it is relatively easy to directly capture the target. Detumbling of space debris actually refers to the process of using external control torque to attenuate the target angular velocity, thereby stabilizing the target attitude. Detumbling of space debris can be divided into contact and noncontact depending on whether the force is in contact with the target. Contact detumbling mainly includes the use of a deceleration brush, mechanical impact, a tether robot, and so on. Noncontact detumbling includes the use of gas, electrostatic force, electromagnetic force, an ion beam, a laser, and other methods, and has the advantages of long operating distance, high safety, and strong attitude tolerance. It is also able to provide multi-degree-of-freedom acting torque, but its impact force is generally in the order of mN [5]. For large space debris with large mass and high speed, noncontact detumbling requires dozens or even hundreds of hours at low efficiency, which causes the energy loss of the device to increase and the number of reuses to be reduced. Contact detumbling provides greater control force, usually reaching the order of N [5]. Among the different methods, mechanical impact provides the largest braking force, which is a very efficient method of active detumbling.

JAXA's Nishida and Kawamoto proposed to eliminate rotation before capture using the elastic contact force between a deceleration brush and the target [6]. Kawamoto, of the national aeronautics and space laboratory of Japan, used alternating attenuation of the target nutation angle and self-rotation velocity by multiple contact pulse forces [7]. Hovell used a viscoelastic rope system attached to the rotating space debris surface and controlled the target speed until its attitude was stable through rope tension and damping force during deformation [8]. Matunaga, of the Tokyo institute of technology, took the elastic ball as the end-effector of a mechanical arm and used the normal impact force and friction force generated by the elastic impact between the mechanical arm and the target surface to attenuate rotation [9]. In the existing research on mechanical impact detumbling, a constant pressure-based impact model established by Matunaga was adopted by many researchers. In order to ensure uniform distribution of the impact force during impact, Matunaga used a gas-filled constant pressure ball as the impact end-effector, but did not consider the change in internal pressure of the gas during impact, which affects the identification accuracy of the impact force. In fact, the internal pressure of a constant pressure ball changes during impact. Matunaga's model therefore contained defects in the impact process, making the identification value of the impact force smaller.

In this paper, aiming to improve the theoretical defect of Matunaga's model in the calculation of impact force, the identification method of impact force considering pressure change during the impact process is proposed, a model of nonlinear impact force is established, and an experimental system with a variable-pressure flexible end-effector is set up. The impact tests under different initial conditions are carried out, and the experimental results are compared with the calculated results of the model. The accuracy of the impact force identification method is verified. The proposed method provides accurate guidance for the design and control of a space debris acquisition manipulator, proving to be of great significance to the application of active mechanical impact detumbling of space debris.

## 2. Concept of the Mechanical Impact Active Detumbling

Mechanical impact active detumbling gradually reduces the residual angular kinetic energy of space debris by applying mechanical impact multiple times and effectively increases the success rate to successfully capture the space debris. The purpose of applying impact to space debris is to reduce its residual angular kinetic energy, therefore, the direction of the mechanical impact should be opposite to the speed of the target point. For example, when the space debris is nutating, the trajectory of the endpoint on the space debris axis forms a circle and the impact direction is along the tangent of the trajectory. The mechanical impact is applied in various ways, such as via flexible objects or robotic end effectors. The process of the mechanical impact active detumbling is shown in Figure 1. In this solution, the mechanical impact is applied by a flexible end-effector.

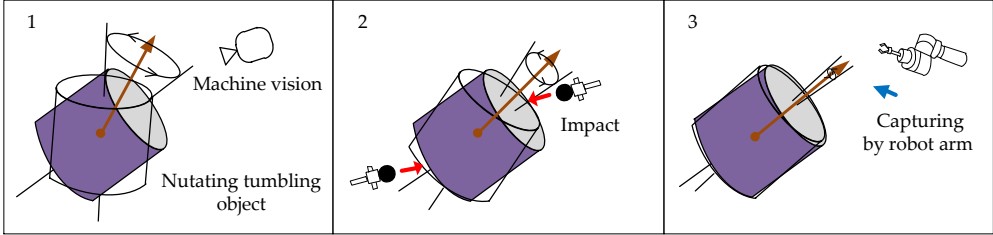

**Figure 1.** Mechanical impact active detumbling.

The process of mechanical impact active detumbling is mainly divided into three phases. In phase 1, the calculation of the impact pose is performed using the identified parameters. Specific impact pose analysis is shown in Figure 2. In phase 2, space debris is impacted by the flexible end-effector, which is controlled by the robot arm. The angular momentum of space debris is reduced by impacting space debris. In phase 3, when the impact is complete, the system of detumbling performs attitude compensation while separating from space debris. These sequences are repeated until the magnitude of the space debris' angular momentum decreases sufficiently. Finally, the space debris is captured by the robot arm.

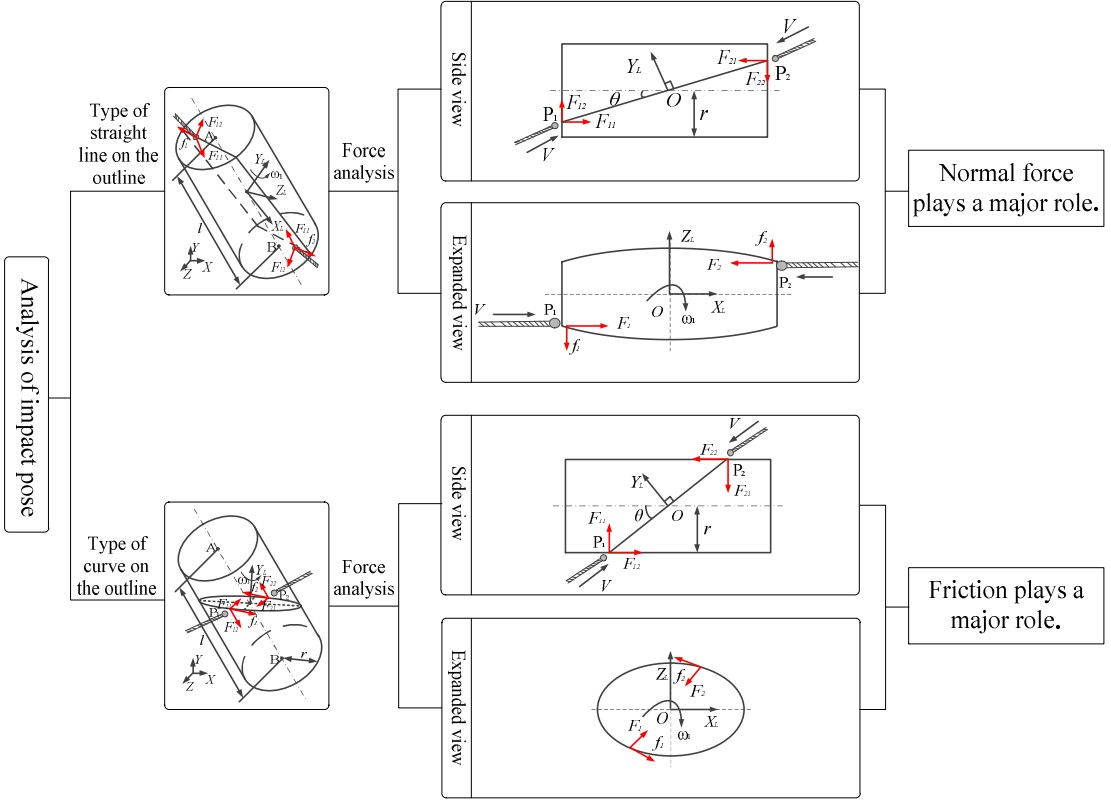

**Figure 2.** Analysis of impact pose.

As seen from Figure 2, the calculation method of the mechanical impact posture effectively solves the problem of unsatisfactory impact results and provides theoretical guidance regarding mechanical impact active detumbling. The precise identification of the impact force plays a vital role in the process of space debris active detumbling, because the choice of impact position depends on the accuracy of the impact force identification.

### 3. The Identification Method of Impact Force

The impact force is applied to the space debris through the collision between the impact end-effector and the space debris, which is an important part of the large space debris active detumbling technology. In this section, the identification method of mechanical impact force is mainly analyzed. Firstly, the impact end-effector is introduced. Inflatable structures are widely used in the field of aerospace because of their light weight, ease of manufacture, and low cost [10–12]. For mechanical impact active detumbling, the use of variable pressure flexible ends provides greater friction and better protection of cooperative targets. Secondly, the modeling process based on the impact force of a variable pressure flexible end-effector is analyzed. Finally, the impact force model under this condition is determined.

*3.1. Flexible End-Effector with Variable Pressure*

The impact model of a constant pressure-based elastic ball was established by Matunaga [9]. In fact, with the change in the amount of deformation during the impact process, the volume of the elastic ball decreases, the internal pressure of the elastic ball increases, and certain parameters such as material and stiffness undergo nonlinear changes which further affect the impact force. Therefore, a flexible end-effector with variable pressure is used, consisting of ethylene-propylene rubber material due to the complex environments found in space, such as harsh temperatures, high vacuum, microgravity, strong radiation, and strong electromagnetic fields. The flexible end-effector and the space robot arm are connected by a flange. The space debris is impacted by control of the robotic arm.

*3.2. Impact Force Modeling*

The modeling process of Matunaga's normal impact force is as follows [9]. The internal pressure $p$ of the impact end-effector is assumed to be constant during the impact time. Thus, the magnitude of the normal impact force $F_n$ can be geometrically derived using the amount of depression of the inflatable ball $\delta$, as follows:

$$F_n = PS = P\pi\{2R_c\delta - \delta^2\}, \tag{1}$$

where $R_c$ is the radius of the model.

In fact, the impact end-effector is deformed during impact, which causes the internal pressure $P'$ of the impact end-effector to increase. The normal impact force should be expressed as $F_n' = P'S$. Because $P'$ is greater than $P$, $F_n'$ is greater than $F_n$, which causes the normal impact force calculated by Matungaga's model to be smaller than the actual value.

The modeling process after considering the internal pressure change of the flexible end-effector during impact is as follows. The object of this study is large space debris. The size of the flexible end-effector is much smaller than the size of the space debris, so the contact surface of the space debris is simplified to a plane. The impact force model is shown in Figure 3, where $k_n$ is the equivalent normal stiffness, $c_n$ is the equivalent normal damping, $k_t$ is the equivalent tangential stiffness, $c_t$ is the equivalent tangential damping, $p$ is the initial pressure, $P'$ is the pressure after the impact, $v_1$ represents the initial impact velocity of the end-effector, and $v_2$ is the component velocity of space debris.

It is assumed that the space debris is rigid and that the flexible end-effector is only elastically deformed. Assumptions are made, wherein (1) the gas is isothermal and (2) the perimeter of the cross-section of the membrane is constant. The cross-section is changed from a circular shape to a racetrack shape, as shown in Figure 4.

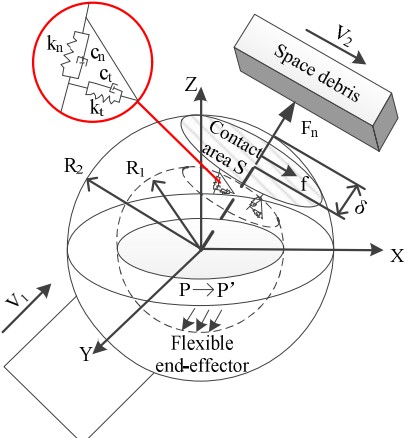

**Figure 3.** Impact force model.

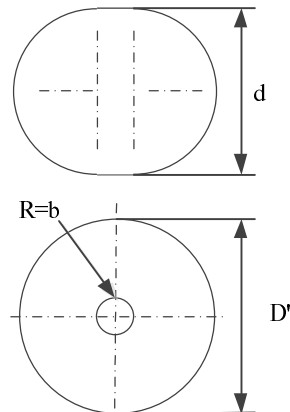

**Figure 4.** Section structure change.

It is assumed that the amount of depression of the inflatable ball under static load is $\delta$, the height of the section after compression is $d = D\text{-}\delta$, and the radius of the contact surface is $b$. $D$ is the diameter of the elastic ball before deformation. $D'$ is the diameter of the elastic ball in the direction perpendicular to the amount of depression after deformation and is also the diameter of the elastic ball after deformation. $d$ represents the height of the elastic ball in the direction of the depression amount after deformation. According to the constant section perimeter, the deformed volume is as follows:

$$V' = 2\pi \int_0^{d/2} \left( b + \sqrt{(d/2)^2 - y^2} \right)^2 dy + \pi b^2 d. \tag{2}$$

Based on the isothermal assumption, the internal pressure after deformation is as follows:

$$P' = \frac{P_0 V_0}{V'} = \frac{P_0 D^3 / 6}{2 \int_0^{d/2} \left( b + \sqrt{(d/2)^2 - y^2} \right)^2 dy + b^2 d}. \tag{3}$$

According to the force-balance relationship between the pressure plate and the contact part of the inflatable ball, the elastic force $F_k$ is as follows:

$$F_k = P' \pi b^2 . \tag{4}$$

According to Hunt Crossley's damping model [13,14], the damping force $F_c$ is as follows:

$$F_c = -c_n F_k \dot{\delta}_n. \tag{5}$$

The damping coefficient is as follows:

$$c_n = \frac{3\left(1 - e^2\right)}{4v_1}, \tag{6}$$

where $e$ is the recovery coefficient of the material and $v_1$ is the initial relative velocity of the impact. In summary, based on Hertzian contact theory [13,14], the normal impact force $F_n$ is as follows:

$$F_n = F_k + F_c = \frac{\pi\delta^2 P_0 D^3/6}{32 \int_0^{d/2}\left(b + \sqrt{(d/2)^2 - y^2}\right)^2 dy + 16b^2 d} - c_n F_k \dot{\delta}_n \tag{7}$$

Different from the friction behavior of metal, the friction coefficient of rubber in the process of friction is not constant. The friction characteristics of rubber can be accurately described by the following formula as [15]:

$$f = F_{adh} + F_{hyst} = \frac{\pi}{4}K\sigma_0\frac{F_n}{p_r}\tan\delta + \frac{\pi F_n}{10}\left(\frac{3p_c}{E^*}\right)^{\frac{2}{3}}\tan\delta, \tag{8}$$

where $F_n$ is the load, $\sigma_0$ is the tensile strength, $K$ is a constant, $P_r$ is the real contact pressure, $P_c$ is the contour contact pressure, $E^*$ is the composite elastic modulus, and $\tan\delta$ is the loss factor. $F_{adh}$ denotes the adhesion friction between the contact surfaces due to the continuous generation and destruction of the adhesion. $F_{hyst}$ denotes the hysteresis friction caused by the periodic deformation of the rubber due to the unevenness of the contact surface.

## 4. Verification Experiment of Impact Force

In order to analyze the impact of the flexible end-effector and space debris, an impact force identification system was designed, as shown in Figure 5, mainly including an impact test bench, an impact force acquisition device, a data acquisition instrument, a computer, and accessory components. The experimental setup used for the entire impact force measurement process is shown in Figure 6. Details of the main equipment are shown in Table 1.

**Table 1.** Details of the main equipment.

| Species | Select | Model | Remark | | |
|---|---|---|---|---|---|
| Signal amplifier | KISTLER multichannel charge amplifier | 5080 A | | | |
| Force gauge | KISTLER piezoelectric force gauge | 9257B | Sensitivity (pC/N) | Fx<br>Fy<br>Fz | −7.5<br>−7.5<br>−3.7 |
| | | | Sampling frequency (HZ) | 1000 | |
| Signal acquisition instrument | KISTLER data acquisition system | 5697A1 | | | |
| Mechanical arm | Eft's ER10L-C10 articulated robot | ER10L-C10 | Possesses 6° of freedom and achieves multi-angle impact. Maximum movement speed of 2 m/s, maximum movement radius of 2022 mm, and repeatable positioning accuracy of ±0.08 mm, thereby meeting the needs of this experiment. | | |

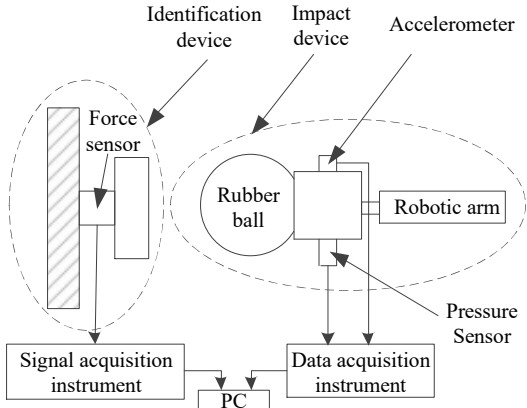

**Figure 5.** Impact force identification system diagram.

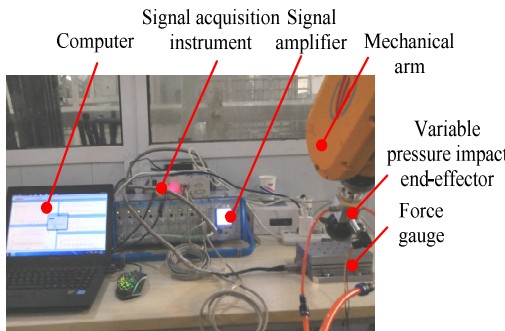

**Figure 6.** Test device of impact.

The flexible end-effector with variable pressure is shown in Figure 7. The parameters of its geometric model are shown in Table 2. By inflating the inside of the sphere to change the internal pressure value, the pressure sensor collects numerical signals in real time for feedback.

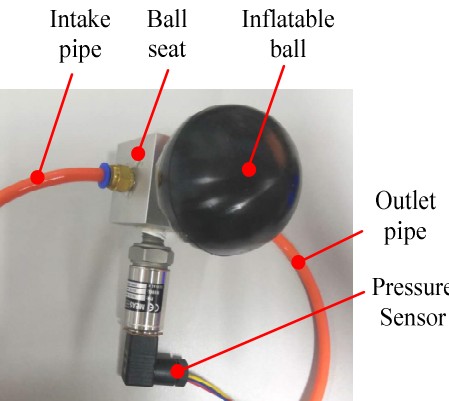

**Figure 7.** The flexible end-effector with variable pressure.

**Table 2.** Geometric model parameters.

| Parameter | Value |
|---|---|
| Inner diameter | 40 mm |
| Outer diameter | 60 mm |
| Poisson's ratio | 0.47 |
| Elastic modulus | 0.0078 GPa |

In the experiment, the single variable principle was used. The movement speed of the robot arm was controlled to 0.4 m/s. The impact time *t* was 0.27 s. According to the depression amount $\delta$ and the pressure $P'$ after the impact, the normal impact force was identified using Equations (2), (3), and (7). The initial pressure of the flexible end-effector was changed to 1.0 MPa, 1.5 MPa, 2.0 MPa, and 2.5 MPa, with the experiments performed in sequence. The experimental results are shown in Figure 8. The peak normal impact forces were 48.519 N, 54.669 N, 62.857 N, and 69.183 N, respectively.

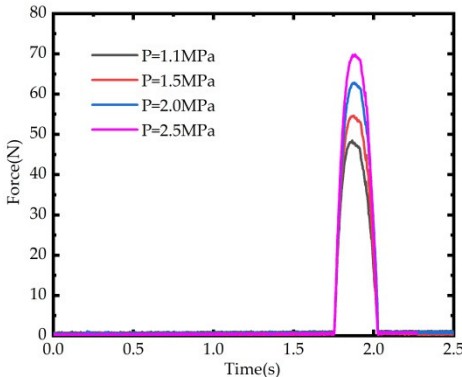

**Figure 8.** Relationship of normal impact force and pressure.

The actual impact force was loaded and unloaded, which was different from the step signal used in the simulation. The pressure at the impact end-effector was achieved through a pneumatic control system, including an intake pipe, a proportional valve, a check valve, an oil mister, a filter, and other parts. At the same time, it was difficult to maintain the same pressure for each experiment due to defects in the manufacturing process causing leakage at the impact end-effector. For this reason, 16 sets of data were measured at each pressure point, and the average value was taken as the normal contact force under the pressure; every effort was made to avoid experimental errors. The comparison of the theoretical and experimental values of the normal impact force is shown in Figure 9. The comparative data are shown in Table 3, where the identification error was less than 6.7%. The rationality of the theoretical calculation was clearly proven by the change trend of the experimental data.

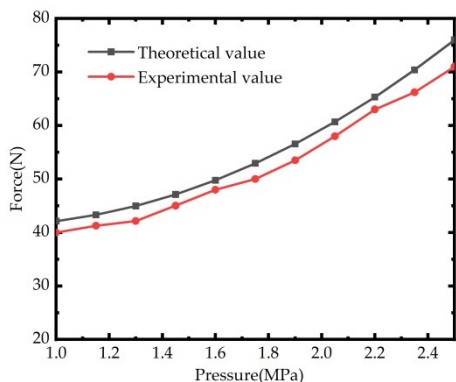

**Figure 9.** Normal impact force comparison results.

**Table 3.** Comparative data of the normal impact force.

| Theoretical Value/N | 42.1 | 43.3 | 44.9 | 47.1 | 49.7 | 52.9 | 56.5 | 60.7 | 65.3 | 70.4 | 75.9 |
|---|---|---|---|---|---|---|---|---|---|---|---|
| Experimental Value/N | 39.8 | 41.3 | 42.1 | 44.8 | 48.2 | 50.3 | 53.5 | 57.9 | 62.6 | 66.2 | 70.8 |
| Identification Error/% | 5.4 | 4.6 | 6.2 | 4.8 | 3.1 | 4.9 | 5.3 | 4.9 | 4.1 | 5.9 | 6.7 |

The friction test device is shown in Figure 10. The initial impact speed was 0.4 m/s and the impact time t was 0.27 s. The inside of the rubber ball was inflated to change the internal pressure by adjusting

the proportional valve. The comparison between the experimental data and the theoretical data of the friction force when the internal pressure was 2.05 Mpa is shown in Figure 11. The actual friction force was greater because the vibration generated during the impact process and the accuracy error of the experimental device had a nonlinear effect on the friction force. The experimental results of sliding friction obtained during the experiment are shown in Figure 12. When the internal pressure was in the range of 1.5–1.9 MPa, the friction force was obviously nonlinear. There are several reasons for this, including because the rubber material is elastic material, generating different degrees of vibration during the experiment as the impact force changed. Stress and strain were not synchronized due to the viscoelasticity of the rubber material. With the change of the internal pressure of the impact end-effector, the contact area changed during the experiment. These reasons are difficult to describe accurately with current theoretical research, and friction is difficult to control. Therefore, this range of pressure when designing flexible end-effectors or performing mechanical impact active detumbling should be avoided. The research purpose of this paper was to build a more accurate mathematical model to describe the impact force, but through numerical calculations and experiments, accurate theoretical predictions and practically controllable pressure ranges were found, providing guidance for practical use in engineering. The experiments showed that the friction coefficient was not constant and the identification error of the friction force was 22.6%, which did not meet the requirements of accuracy.

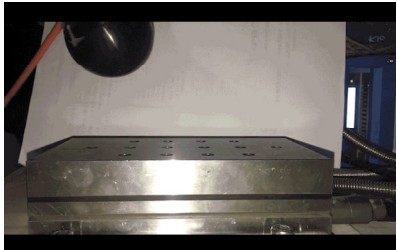

**Figure 10.** Friction test.

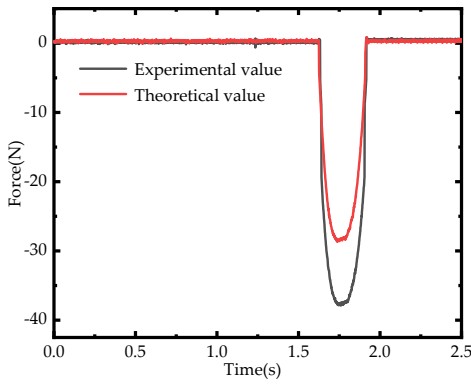

**Figure 11.** Comparison of experimental friction.

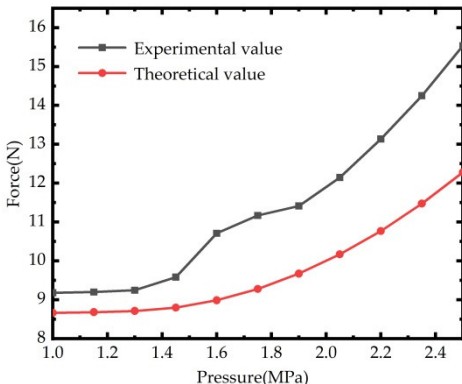

**Figure 12.** Friction force comparison results.

The experiments showed that the rubber ball slipped during the sliding process, causing its volume to deform, as shown in Figure 13. The red dashed line in Figure 13 represents the slipping process. In the sliding process, a frictional force $f_{disp}$ was generated, which prevented slipping. The $f_{disp}$ is as follows:

$$f\text{disp} = (a + a\exp(bs)[c\sinh(ds) - e\cosh(ds)])F_n = \mu F_n, \tag{9}$$

where $s$ is the sliding displacement, $F_n$ is the load, and $\mu$ is the coefficient. Each coefficient was fitted using experimental data as

$$\begin{cases} a = 0.74769 - 0.01684 F_n \\ b = -0.13056 - 0.01728 F_n + 5.35714 exp^{-4} F_n^2 \\ c = -1.04239 + 0.00583 F_n - 4.07238 exp^{-4} F_n^2 \\ d = 0.010619 + 0.01829 F_n - 7.75 exp^{-4} F_n^2 \\ e = 2.69364 - 0.05815 F_n + 0.00712 exp^{-4} F_n^2 \end{cases} \tag{10}$$

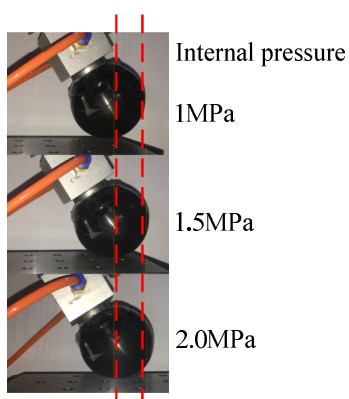

**Figure 13.** Results of sliding deformation.

In summary, the corrected friction is as follows:

$$f = \frac{\pi}{4} K \sigma_0 \frac{F_n}{p_r} \tan \delta + \frac{\pi F_n}{10} \left( \frac{3p_c}{E^*} \right)^{\frac{2}{3}} \tan \delta + \mu F_n. \tag{11}$$

The comparison of the actual value, the theoretical value, and the corrected value of the friction force are shown in Figure 14. The comparative data are shown in Table 4. The identification error of the friction force was less than 6.9%.

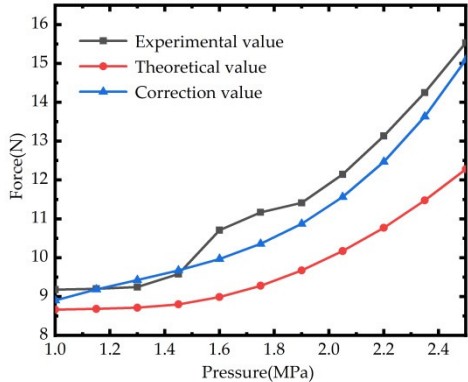

**Figure 14.** Corrected friction force comparison results.

**Table 4.** Comparative data for the friction force.

| Experimental Value/N | 9.18 | 9.2 | 9.25 | 9.58 | 10.7 | 11.1 | 11.4 | 12.1 | 13.1 | 14.3 | 15.5 |
|---|---|---|---|---|---|---|---|---|---|---|---|
| Correction Value/N | 8.91 | 9.18 | 9.42 | 9.67 | 9.97 | 10.4 | 10.8 | 11.6 | 12.6 | 13.8 | 15.2 |
| Identification Error/% | 2.9 | 0.2 | 1.8 | 0.1 | 6.9 | 6.3 | 5.2 | 4.1 | 3.8 | 3.4 | 3.1 |

The corrected friction force considered the large deformation of the impact ball structure during sliding, which resulted in a larger contact area and was also in line with the calculation method of the actual friction force. Therefore, the correction value of the friction force was larger than the initial theoretical value and was closer to the actual theoretical calculated value. The numerical trends agreed with these findings.

## 5. Simulation of Active Detumbling

In this paper, the Euler angle pose description method was adopted to describe the motion state of space debris. The last stage of the rocket was selected for dynamic modeling of the space debris, providing a dynamic model for the simulation of the subsequent detumbling method. In order to describe the motion status of space debris in orbit, the following assumptions need to be made for space debris:

(1) Space debris is in geosynchronous orbit. When the satellites used for detumbling reach the designated position, the relative position between them can be considered to not change.
(2) The space debris is a regular cylinder similar to the last stage of the rocket, with its center is located on the orbital line and the center point not drifting during movement.
(3) The rotational inertia of space debris is concentrated on the three main inertia axes and the rotational inertia in other directions is relatively small.
(4) The space debris is a rigid body and does not deform during normal operation and detumbling.

In active detumbling of space debris, the detumbling satellite observes the target movement through the visual sensor. Then, the surface state of the space debris is analyzed to obtain the topography information of the position where the space debris can be impacted and the target contact area is selected. Finally, the impact position and impact force are calculated by combining the motion state of the space debris and the morphological characteristics of the impactable surface, allowing the final impact posture and impact moment to be determined. Figure 15 shows the execution process of mechanical impact active detumbling.

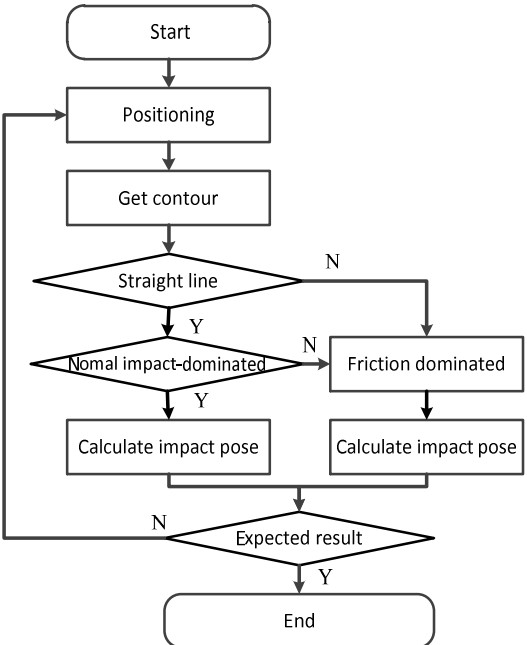

**Figure 15.** The execution process of mechanical impact active detumbling.

This simulation used the rocket's last stage as the object of active detumbling, which was approximated as a regular cylinder with three-axis angular momentum and lost attitude adjustment capability. In this simulation experiment, the model had a radius of 1000 mm, a length of 3000 mm, and a mass of 1000 kg. The dynamic model of space debris is shown in Figure 16. The symbols in Figure 16 are defined as follows: $OXYZ$ is the base coordinate system, $OX_LY_LZ_L$ is the angular momentum coordinate system, $A$ and $B$ are the center points of the upper and lower end faces of the model, $P_1$ and $P_2$ are the impact positions, $l$ is the length of the model, and $r$ is the radius. The $Y_L$ axis of the angular momentum coordinate system in Figure 16 is the direction of the combined angular momentum. The motion state of large space debris was free tumbling motion with a changing nutation angle. The trajectory of the center point A of the space debris end-face in the base coordinate system is shown in Figure 17.

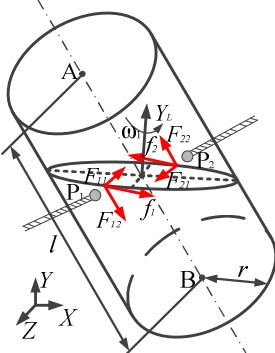

**Figure 16.** Dynamic model of space debris.

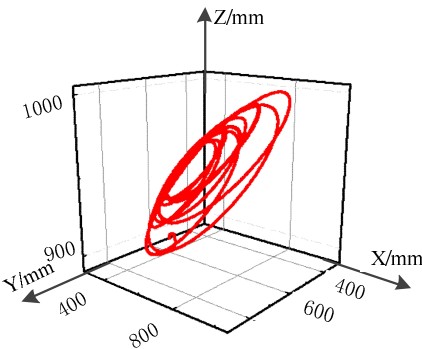

**Figure 17.** Endpoint trajectory graph.

In this simulation, the impact time was 0.4 s, the number of simulation steps was 10,000 steps, the time was 2000 s, and the step size was 0.2 s. Natural disturbance forces and moments were ignored during the simulation. The mechanism of detumbling in this paper involved the direction of the moment of force formed by the impact force being the direction of the combined angular momentum, thereby weakening the combined angular momentum of the space debris.

According to the relationship between the angle between the space debris axis and the combined angular momentum and the shape of the contour line, space debris can be divided into the following two cases:

(1) The change in the space debris' angular momentum after the impact when there were straight lines on the contour of the space debris is shown in Table 5. The precession angular velocity of the space debris after the impact was greatly reduced, as shown in Figure 18. The curve of nutation angle before and after the impact is shown in Figure 20.

(2) The change in the space debris' angular momentum after the impact when there was no straight line on the contour of the space debris is shown in Table 6. The precession angular velocity of the space debris after the impact was reduced, as shown in Figure 19. The curve of the nutation angle before and after the impact is shown in Figure 21.

**Table 5.** Comparative data for the angular momentum of space debris.

| Axis | Pre-Impact Value(N·mm·s) | Post-ImpactValue (N·mm·s) | Reduction Rate |
|---|---|---|---|
| X | 24,764 | 9137 | 63.10% |
| Y | −41,782 | −15,407 | 63.13% |
| Z | 54,270 | 19,954 | 63.23% |
| Total angular momentum | 72,829 | 26,815 | 63.18% |

**Table 6.** Comparative data for the angular momentum of space debris.

| Axis | Pre-Impact Value (N·mm·s) | Post-ImpactValue (N·mm·s) | Reduction Rate |
|---|---|---|---|
| X | 24,768 | 18,001 | 27.32% |
| Y | −41,789 | −30,321 | 27.44% |
| Z | 54,263 | 39,344 | 27.49% |
| Total angular momentum | 72,829 | 52,834 | 27.45% |

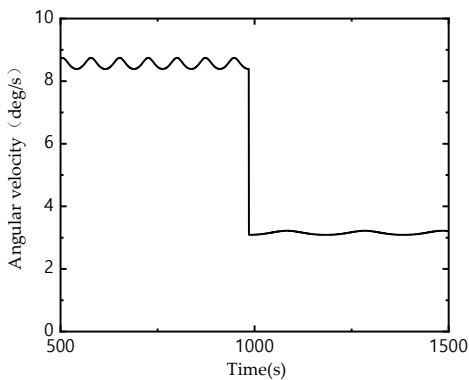

**Figure 18.** Precession angular velocity.

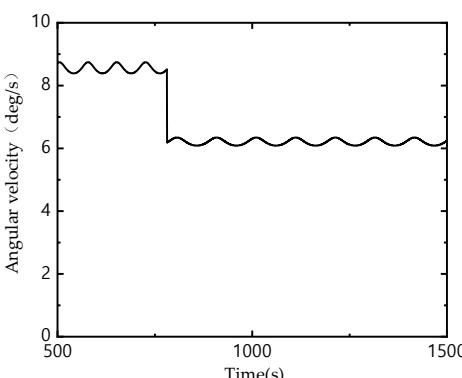

**Figure 19.** Precession angular velocity.

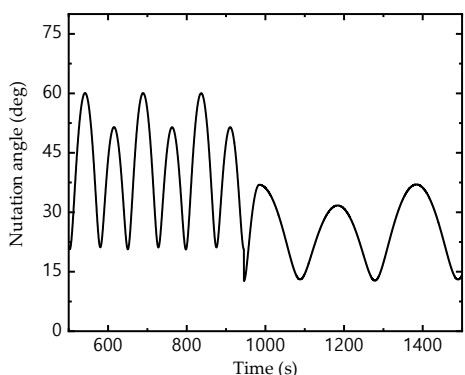

**Figure 20.** Change curve of the nutation angle.

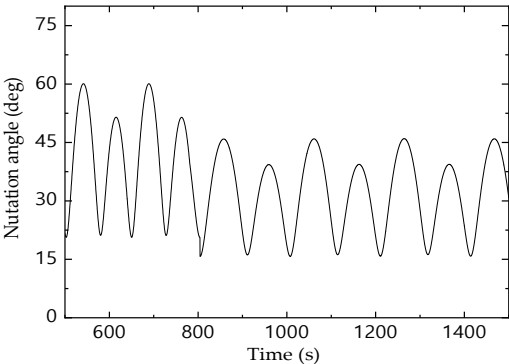

**Figure 21.** Change curve of the nutation angle.

The data in Table 5 show that the triaxial angular momentum of the space debris after the impact in Case (1) was attenuated by more than 63%. The data in Table 6 indicate that the angular momentum of the three axes of the space debris after the impact in Case (2) was attenuated by more than 27%. As can be seen from Figures 18 and 19, the precession angular velocity was greatly reduced after impact in both cases. From Figures 20 and 21, it is clearly observed that due to the decrease of the precession angular velocity, the cycle of the nutation angle of the space debris became longer and the nutation angle was reduced after impact. In both cases, the trend of the angular velocity of space debris before and after impact is shown in Figure 22, showing that mechanical impact active detumbling effectively attenuated the triaxial angular momentum of space debris, but with different levels. Therefore, the effect of mechanical impact active detumbling was successfully proven.

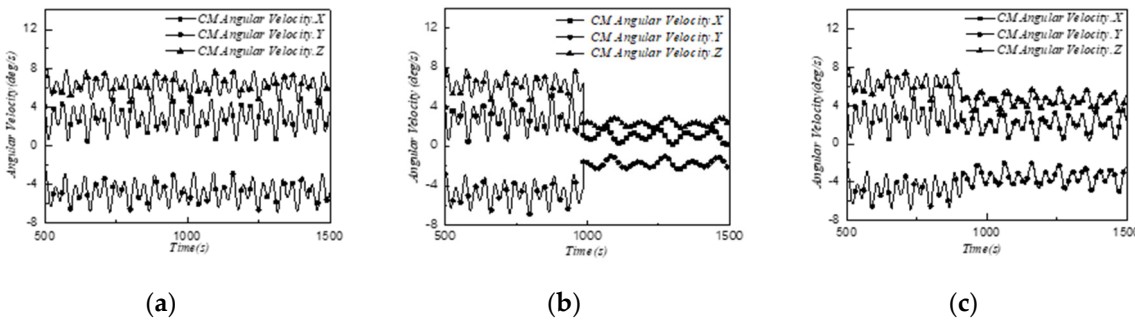

(**a**)　　　　　　　　　　(**b**)　　　　　　　　　　(**c**)

**Figure 22.** The trend of triaxial angular velocity. (**a**) Before impact; (**b**) Case (1) after impact; (**c**) Case (2) after impact.

Taking Case (2) as an example, the changing trend of space debris after impact was demonstrated. Figure 23 shows a comparison of space debris before and after impact. Figure 24 shows a comparison of the two-dimensional displacement of space debris before and after impact. The effect of mechanical impact active detumbling is directly demonstrated in Figures 23 and 24. The effect of active detumbling was therefore proven once again.

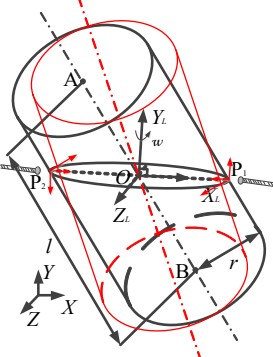

**Figure 23.** The comparison of space debris.

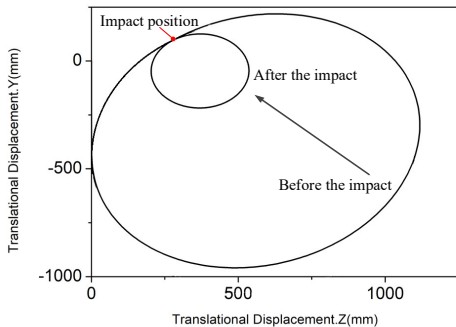

**Figure 24.** The displacement of space debris.

This simulation verified that the impact reduces the triaxial angular momentum of space debris simultaneously and the direction of the combined angular momentum does not change during impact, reflecting the high efficiency of mechanical impact active detumbling. The efficiency of normal force is much greater than that of friction force. In this paper, the feasibility of mechanical impact active detumbling was verified through simulation experiments, thereby laying the foundation for mechanical impact active detumbling research.

## 6. Conclusions

In this paper, the variable pressure flexible end-effector was taken as the main research object. The model of the impact force between the flexible end-effector and the space debris was established and an impact force identification method considering pressure changes was proposed. Formulae regarding the normal impact force and tangential friction force of a flexible end-effector with variable pressure were derived and impact experiments with one degree of freedom were carried out. The accuracy of the impact force model was verified. The simulation results and experiments showed that the impact force model had a strong nonlinear relationship. The identification error of normal impact force was less than 6.7% and the identification error of tangential friction force was less than 6.9%, thereby meeting the requirements for detumbling technology of space debris.

**Author Contributions:** Methodology, Z.W. and H.Z.; validation, B.Z., X.L., and R.M.; writing—original draft, Z.W.; writing—review and editing, R.M. All authors have read and agreed to the published version of the manuscript.

**Funding:** This research was supported by the National Natural Science Foundation of China under grant 51705128. This research was also funded by the Key Laboratory of Space Intelligent Control Technology Open Fund, grant number ZDSYS-2017-08. This research was funded by the Hebei Province Higher Education Science and Technology Research Project, grant number BJ2019049. This research was funded by the Natural Science Foundation of Hebei Province, grant number E2019202132, E2019202169. This research was funded by the Natural Science Foundation of Tianjin, China, grant number 18JCTPJC54700, 19JCYBJC19100. This research was supported by the National Key Laboratory of Science and Technology on Vacuum Technology and Physics, Lanzhou Institute of Physics (ZWK1803).

**Acknowledgments:** This work was supported by the National Natural Science Foundation of China (51705128), the Key Laboratory of Space Intelligent Control Technology Open Fund (ZDSYS-2017-08), the Hebei Province Higher Education Science and Technology Research Project (BJ2019049), the Natural Science Foundation of Hebei Province (E2019202132, E2019202169), the Natural Science Foundation of Tianjin, China (18JCTPJC54700, 19JCYBJC19100) and the National Key Laboratory of Science and Technology on Vacuum Technology and Physics, Lanzhou Institute of Physics (ZWK1803).

**Conflicts of Interest:** The authors declare no conflict of interest.

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
