# Peer review of "Impact Force Identification of the Variable Pressure Flexible Impact End-Effector in Space Debris Active Detumbling"

_applsci, doi:10.3390/app10093011_

Round 1
Reviewer 1 Report
Dear Authors,
Please, find attached the below review.

Author Response
Response to Reviewer 1 Comments
Manuscript ID: applsci-734121
Type of manuscript: Article
Title: Impact force identification of the variable pressure flexible impact
end-effector in space debris active detumbling
Authors: Ziying Wei, Huibo Zhang, Baoshan Zhao, Xiaoang Liu *, Rui Ma *
Dear Editors and Reviewers:
We would like to thank the editor and the reviewers for their careful evaluation and valuable suggestions for our revision. Those comments are all valuable and very helpful for revising and improving our paper, as well as the important guiding significance to our researches. We have studied comments carefully and have made correction which we hope meet with approval. Revised portion are marked in red in the paper. The main corrections in the paper and the responds to the reviewer’s comments are as flowing:
Responds to the reviewer #1
1.(lines 6-7) In the description of the institutions given by the authors as an affiliate - point 1, Mr. Xiaoang Liu's web address is missing. (lines 8-9) Whereas in the second description - item 2 Tianjin Key (...), the affiliate institution address is missing.
Reply: We are very sorry for our negligence. We added this part according to the Reviewer’s suggestion, please see: Page 1, Line 7-10.
2.(between lines 116 and 117) Chapter 3, before point 3.1. The flexible (...) is missing in the chapter introduction - a few sentences.
Reply: We are very sorry for our negligence. We added this part according to the Reviewer’s suggestion, please see: Page 4, Part 3, Line117 -125.
3.(line 157) Not all mathematical formulas have given sources from the list of references of the article- e.g. (4).
Reply: We are very sorry for our negligence. We added this part according to the Reviewer’s suggestion , please see: Page 6, Line 169,172,175 .
4.(lines 131 and 132) No mathematical formula number was given. Also, the meaning of the "R" symbol is not given.
Reply: We are very sorry for our negligence. We added this part according to the Reviewer’s suggestion, please see: Page 4, Part 3, Line 140-141.
4.Please check that all symbols given in the mathematical formulas are explained/have the meaning given in the text and try to supplementing of missing meanings - e.g.
Reply: We are very sorry for our negligence. We added this part according to the Reviewer’s suggestion, please see: Line 152,163-164,246-247.
5.(line 175) Figure 5. The title is not fully adequate to its description in the text (line 170 - 171).
Reply: We are very sorry for our negligence. We added this part according to the Reviewer’s suggestion, please see: Page 4, Part 3, Line 183, 188.
6.(line 202) Figure 10 requires a broader description (1-2 sentences), without a description that is illegible.
Reply: We are very sorry for our negligence. We added this part according to the Reviewer’s suggestion, please see: Page 8, Part 4, Line 219-221.
7(lines 229-234) I believe that the individual points should be given as follows:
(....) the following two cases:
1.When there are (...)
2.When there is (...)
Reply: We are very sorry for our negligence. We added this part according to the Reviewer’s suggestion, please see: Page 12, Part 5, Line 285,289.
8.(line 236) There is no description on the abscissa of the coordinate system (Fig. 14.).
Reply: We are very sorry for our negligence. We have added "Time" to the abscissa of the coordinate system , please see: Page 12, Part 5, Line 294.
9.(lines 222-247) Authors did not refer in the text to the digital data from the presented diagrams and tables when describing part of the results of their research - Chapter 5 - Simulation of active detumbling. it is not about providing all values, but supporting the conclusions with specific data - as was done in chapter 4 - line 213.
Reply: We are very sorry for our negligence. As Reviewer suggested that we should support the conclusions with specific data . So, we added this part according to the Reviewer’s suggestion, please see: Page 12, Part 5, Line 299-309.
10.(lines 270 and 290). Authors' contribution - it should be made according to the template on the publisher's website. It needs correction.
Reply: We are very sorry for our negligence. We have modified this part according to the Reviewer’s suggestion, please see: Page 13,14,, Part 7, Line 341-373.
Special thanks to you for your good comments.
We tried our best to improve the manuscript and made some changes in the manuscript. These changes will not influence the content and framework of the paper. And here we did not list the changes but marked in red in revised paper.
We appreciate for Editors/Reviewers’ warm work earnestly, and hope that the correction will meet with approval.
Once again, thank you very much for your comments and suggestions.
Sincerely yours,
Ziying Wei, Huibo Zhang, Baoshan Zhao, Xiaoang Liu *, Rui Ma *

Reviewer 2 Report
Dear Authors
The paper is interesting, but there are a few questions devoted to the methodology, theory and experiments.
First of all, you should perform experiments and ask yourself about the obtained data and it's further interpretation and mathematical description by analytical formulas.
The experimental test rig is not described in details, the information about data acquisition system, sensor characteristics, real-time mode, etc, are not given.
From Fig. 9 we can learn that the proposed theory is not adequate or the experimental test rig is not functioning properly.
This is important,with respect to the result presented in Fig. 11 and 13.
Let's assume that the test rig works well. Therefore it is an request to analyse the physic of the phenomena and reformulate the model, to fit the experimental data properly.
What about the contact force, Stribeck effect, Coulomb friction, impact force, etc? Is the experiment repeatable? Could you show a few results?
Especially nonlinear and non monotonous effect for pressures in the range of 1.5÷1.9MPa must be considered.
Finally, the simulation of the detumbling is given is parts. According to the theory presented in chapters 2 and 3 ought to be confirmed and discussed in chapter 5. I wish to see a complete state of the detumbled element before and after force application - all displacements, angles and their velocities before and after detumbling action.
There are a few typos, extra spaces, missing spaces, dots, comas. It is easy to find them out.
A list of references should be extended.
Author Response
Responses to the reviewers' comments
Manuscript ID: applsci-734121
Type of manuscript: Article
Title: Impact force identification of the variable pressure flexible impact
end-effector in space debris active detumbling
Authors: Ziying Wei, Huibo Zhang, Baoshan Zhao, Xiaoang Liu *, Rui Ma *
Dear Editors and Reviewers:
We would like to thank the editor and the reviewers for their careful evaluation and valuable suggestions for our revision. Those comments are all valuable and very helpful for revising and improving our paper, as well as the important guiding significance to our researches. We have studied comments carefully and have made correction which we hope meet with approval. Revised portion are marked in red in the paper. The main corrections in the paper and the responds to the reviewer’s comments are as flowing:
The experimental test rig is not described in details, the information about data acquisition system, sensor characteristics, real-time mode, etc, are not given.
Reply: The description of the experimental test rig is mainly shown in Figs. 5 and 6. Details of the main equipment are shown in Table 1.
From Fig. 9 we can learn that the proposed theory is not adequate or the experimental test rig is not functioning properly.
Reply: This is caused by the experimental test rig. The main reasons are as follows: the pressure at the impact end-effector is achieved through a pneumatic control system. This system includes the intake pipe, proportional valve, check valve, oil mister, filter and other parts. It is difficult to maintain the same pressure for each experiment due to defects in the manufacturing process which can cause leakage at the impact end-effector. Therefore, we measure 16 sets of data at each pressure point, and finally use the average value as the normal contact force under pressure, and make every effort to avoid experimental errors. The rationality of the theoretical calculation can be proved by the change trend of experimental data.
Figure 1. The normal impact force.
Figure 1 shows the different results of measuring normal impact forces under the same conditions. In the experiment, the normal impact force fluctuates. In future experiments, we will continue to improve the experimental system and improve the control accuracy. In order to ensure that the topic of the paper is clear and concise, the data processing process is not listed in detail, but the experimental processing steps are described in text in the experiment.
This is important,with respect to the result presented in Fig. 11 and 13.
Let's assume that the test rig works well. Therefore it is an request to analyse the physic of the phenomena and reformulate the model, to fit the experimental data properly.
What about the contact force, Stribeck effect, Coulomb friction, impact force, etc? Is the experiment repeatable? Could you show a few results?
Reply: The contact force consists of normal impact force and friction. The experimental data of normal impact and friction are shown in Figure 8 and Figure 9, respectively.
Especially nonlinear and non monotonous effect for pressures in the range of 1.5÷1.9MPa must be considered.
Reply: When the internal pressure is in the range of 1.5-1.9MPa, the friction force is obviously non-linear. There are several reasons for this. Because the rubber material is a highly elastic material, different degrees of vibration will be generated during the experiment as the impact force changes. Stress and strain are not synchronized due to the viscoelasticity of the rubber material. With the change of the internal pressure of the impact end-effector, the contact area changed during the experiment. These reasons are difficult to describe accurately with current theoretical research, and friction is also difficult to control. Therefore, we recommend to try to avoid using this range of pressure when the flexible end-effector is designed or the mechanical impact-type active detumbling is performed. The research purpose of this paper is to build a more accurate mathematical model to describe the impact force. On the other hand, we also want to find theoretically accurate and practically controllable pressure ranges through numerical calculations and experiments to provide guidance for practical use in engineering.
Finally, the simulation of the detumbling is given is parts. According to the theory presented in chapters 2 and 3 ought to be confirmed and discussed in chapter 5. I wish to see a complete state of the detumbled element before and after force application - all displacements, angles and their velocities before and after detumbling action.
Reply: According to the reviewer's suggestions , we discussed and added this part, please see: Page 11,12,13, Part 5, Line 268-281, 299-312.
There are a few typos, extra spaces, missing spaces, dots, comas. It is easy to find them out.
Reply: We have made correction according to the Reviewer’s comments. We are very sorry for our incorrect writing in this paper. We have tried our best to correct these problems. If there is also any shortage, please correct.And here we did not list the changes but marked in revised paper.
A list of references should be extended.
Reply: We have made correction according to the Reviewer’s comments, please see: Pages 13and14, Part 7, Line 341-373.
Special thanks to you for your good comments.
We tried our best to improve the manuscript and made some changes in the manuscript. These changes will not influence the content and framework of the paper. And here we did not list the changes but marked in red in revised paper.
We appreciate for Editors/Reviewers’ warm work earnestly, and hope that the correction will meet with approval.
Once again, thank you very much for your comments and suggestions.
Sincerely yours,
Ziying Wei, Huibo Zhang, Baoshan Zhao, Xiaoang Liu *, Rui Ma *

Round 2
Reviewer 2 Report
page 3, row 109, The Fig. 2 presents two impact elements, while the whole paper is devoted to a single impact actuator. It introduces confusion of the reader.
page 4, row 137 add space before [9]
page 6, row 186 not necessary, as well 189, two Figs. located next to each other is correct. Forming as a Table is exaggerated.
page 6, row 169, 172 - put space as follows: 13, 14
page 6, row 175 - add ":" after [15]
page 8, row 219 "Control the robot arm to slide on the contact surface." - unclear and unconnected sequence with the remaining text
page 9, more references and discussion about friction model are requested. We can find that this problem is critical due to the low obtained accuracy (error 22.6% - page 9, row238) as authors stated!
page 11, row 275, I recommend to add two Figures: Fig 17 presenting the model and location of the actuator when the touch/impact is done, Fig. 18 - orbit after impact
It will be easy, because time diagrams are plotted now. Such presentation will help to analyze the simulation and observe results.
page 12, row. 294, Add grid lines for better Figs reading.
page 12. Why the nutation angle is changing frequency only? At the first page (row 64) the reader is expected that that the nutation angle will be minimized. Moreover, according to the schematic diagram presented in Fig. 1. the goal is to damp vibrations.
Author Response
Dear Reviewer:
We would like to thank the editor and the reviewers for their careful evaluation and valuable suggestions for our revision. Those comments are all valuable and very helpful for revising and improving our paper, as well as the important guiding significance to our researches. We have studied comments carefully and have made correction which we hope meet with approval. Revised portion are marked in red in the paper. The main corrections in the paper and the responds to the reviewer’s comments are as flowing:
page 3, row 109, The Fig. 2 presents two impact elements, while the whole paper is devoted to a single impact actuator. It introduces confusion of the reader.
Reply: There are two impact elements. A moment of force is formed by impact at two points. The impact characteristics of a single impact actuator have been studied in order to be able to generate a controllable moment of force.
page 4, row 137 add space before [9]
Reply: We are very sorry for our negligence. We added this part according to the Reviewer’s suggestion, please see: Page 4, Row137.
page 6, row 186 not necessary, as well 189, two Figs. located next to each other is correct. Forming as a Table is exaggerated.
Reply: We are very sorry for our negligence. We added this part according to the Reviewer’s suggestion, please see: Page 7, Row 190.
page 6, row 169, 172 - put space as follows: 13, 14
Reply: We are very sorry for our negligence. We added this part according to the Reviewer’s suggestion, please see: Page 6, Row 169, 172.
page 6, row 175 - add ":" after [15]
Reply: We are very sorry for our negligence. We added this part according to the Reviewer’s suggestion, please see: Page 6, Row 175.
page 8, row 219 "Control the robot arm to slide on the contact surface." - unclear and unconnected sequence with the remaining text
Reply: We are very sorry. We have modified this part according to the Reviewer’s suggestion, please see: Page 8, Row 219.
page 9, more references and discussion about friction model are requested. We can find that this problem is critical due to the low obtained accuracy (error 22.6% - page 9, row238) as authors stated!
Reply: We are very sorry for our negligence.The frictional error of 22.6% was found through multiple experiments. This is due to slip deformation in the rubber. Rubber materials are complex and highly non-linear. Its friction coefficient changes with friction deformation. However, this part of the friction needs to be fitted experimentally because there is no specific model that can accurately describe it. The reference material for the friction test is as follows:
[1] Chen, J H. Study on friction properties of rubber-metal sliding interfa ces in rubber processing. Dissertation for the Master Degree in Engineering, Harbin China, Harbin Institute of Technology,2016.
[2] Chang, JJ. The Research on Friction Characteristic of Banknotes A rithmometer Rubber Roll [C]. Tribology Branch of Chinese Mechanical Engineering Society. Proceedings of the 8th National Tribology Congress, 2007, 309-312.
[3] Zhuang, Y. Study on Dynamic Friction Property of Tire And Its Effect on Vehicle Handling. Dissertation for the Master Degree in Engineering, Changchun China, Jilin University, 2004.
[4] Guo, K H.; Zhuang, Y.; et al. Study on Rubber Friction Test of Automobile Tire. Chinese Journal of Mechanical Engineering, 2004,(10),175-180.
[5] Wang, G Y. Friction and Testing of Rubbers. Special Purpose Rubber Products, 2000, (03), 55-62.
page 11, row 275, I recommend to add two Figures: Fig 17 presenting the model and location of the actuator when the touch/impact is done, Fig. 18 - orbit after impact
It will be easy, because time diagrams are plotted now. Such presentation will help to analyze the simulation and observe results.
Reply: We are very sorry for our negligence. We added this part according to the Reviewer’s suggestion, please see: Page 13, Row 313-319.
page 12, row. 294, Add grid lines for better Figs reading.
Reply: We are very sorry for our negligence. We added this part according to the Reviewer’s suggestion, please see: Page 12, Row 293.
page 12. Why the nutation angle is changing frequency only? At the first page (row 64) the reader is expected that that the nutation angle will be minimized. Moreover, according to the schematic diagram presented in Fig. 1. the goal is to damp vibrations.
Reply: We are very sorry for our negligence. We are also sorry for the fact that the figure on the nutation angle in the text is wrong. Therefore, we have corrected this problem in the article, please see: Page 12, Row 294, 297, 305.
Special thanks to you for your good comments.
We tried our best to improve the manuscript and made some changes in the manuscript. These changes will not influence the content and framework of the paper. And here we did not list the changes but marked in red in revised paper.
We appreciate for Editors/Reviewers’ warm work earnestly, and hope that the correction will meet with approval.
Once again, thank you very much for your comments and suggestions.
Sincerely yours,
Ziying Wei, Huibo Zhang, Baoshan Zhao, Xiaoang Liu *, Rui Ma *

Round 3
Reviewer 2 Report
Dear Authors
Now, the paper sound better than in previous versions. Some editorial bugs are still present. For example: p4.r. 141 Where -> where
Regards
Author Response
Responses to the reviewer's comments
Manuscript ID: applsci-734121
Type of manuscript: Article
Title: Impact force identification of the variable pressure flexible impact end-effector in space debris active detumbling
Authors: Ziying Wei, Huibo Zhang, Baoshan Zhao, Xiaoang Liu *, Rui Ma *
Dear Editors and Reviewers:
We would like to thank the editor and the reviewers for their careful evaluation and valuable suggestions for our revision. Those comments are all valuable and very helpful for revising and improving our paper, as well as the important guiding significance to our researches. We have studied comments carefully and have made correction which we hope meet with approval. Revised portion are marked in yellow in the paper. The main corrections in the paper and the responds to the reviewer’s comments are as flowing:
Now, the paper sound better than in previous versions. Some editorial bugs are still present. For example: p4.r. 141 Where -> where
Reply: We are very sorry for our negligence. We have made correction according to the Reviewer’s comments. We are very sorry for our incorrect writing in this paper. We have tried our best to correct these problems. If there is also any shortage, please correct. And here we did not list the changes but marked in revised paper.
Special thanks to you for your good comments.
We tried our best to improve the manuscript and made some changes in the manuscript. These changes will not influence the content and framework of the paper. And here we did not list the changes but marked in yellow in revised paper.
We appreciate for Editors/Reviewers’ warm work earnestly, and hope that the correction will meet with approval.
Once again, thank you very much for your comments and suggestions.
Sincerely yours,
Ziying Wei, Huibo Zhang, Baoshan Zhao, Xiaoang Liu *, Rui Ma *
